# Ectopic Expression of Grapevine Gene *VaRGA1* in *Arabidopsis* Improves Resistance to Downy Mildew and *Pseudomonas syringae* pv. *tomato* DC3000 But Increases Susceptibility to *Botrytis cinerea*

**DOI:** 10.3390/ijms21010193

**Published:** 2019-12-27

**Authors:** Shanshan Tian, Xiangjing Yin, Peining Fu, Wei Wu, Jiang Lu

**Affiliations:** Center for Viticulture and Enology, School of Agriculture and Biology, Shanghai Jiao Tong University, Shanghai 200240, China; a1083290770@sjtu.edu.cn (S.T.); yinxiangjingsmile@163.com (X.Y.); fupeining@sjtu.edu.cn (P.F.); wuwei_hello@sjtu.edu.cn (W.W.)

**Keywords:** NBS-LRR, *VaRGA1*, disease resistance, histochemical staining, signaling pathways, broad-spectrum

## Abstract

The protein family with nucleotide binding sites and leucine-rich repeat (NBS-LRR) in plants stimulates immune responses caused by effectors and can mediate resistance to hemi-biotrophs and biotrophs. In our previous study, a Toll-interleukin-1(TIR)-NBS-LRR gene cloned from *Vitis amurensis* “Shuanghong”, *VaRGA1*, was induced by *Plasmopara viticola* and could improve the resistance of tobacco to *Phytophthora capsici*. In this study, *VaRGA1* in “Shuanghong” was also induced by salicylic acid (SA), but inhibited by jasmonic acid (JA). To investigate whether *VaRGA1* confers broad-spectrum resistance to pathogens, we transferred this gene into *Arabidopsis* and then treated with *Hyaloperonospora arabidopsidis* (*Hpa*), *Botrytis cinerea* (*B. cinerea*), and *Pseudomonas syringae* pv. *tomato* DC3000 (*Pst*DC3000). Results showed that *VaRGA1* improved transgenic *Arabidopsis thaliana* resistance to the biotrophic *Hpa* and hemi-biotrophic *Pst*DC3000, but decreased resistance to the necrotrophic *B. cinerea*. Additionally, qPCR assays showed that *VaRGA1* plays an important role in disease resistance by activating SA and inhibiting JA signaling pathways. A 1104 bp promoter fragment of *VaRGA1* was cloned and analyzed to further elucidate the mechanism of induction of the gene at the transcriptional level. These results preliminarily confirmed the disease resistance function and signal regulation pathway of *VaRGA1*, and contributed to the identification of R-genes with broad-spectrum resistance function.

## 1. Introduction

Grape is an economically important crop cultivated worldwide. However, its yield and quality are restricted by various pathogens [1,2]. Repeated use of fungicides not only increases production costs and causes environmental pollution, but also leads to the drug resistance of pathogens, which may eventually cause sudden pathogen outbreaks in the future [3,4]. As an alternative to chemical control, planting resistant varieties is cost-effective and environmentally friendly [5]. Utilizing disease-resistant resources of wild grape has become one of the most economical and effective methods to improve the resistance of cultivated varieties [6,7].

Plants generally have two immune system levels. First, transmembrane pattern recognition receptors are used to identify slowly evolving pathogen pattern molecules such as flagellin. Second, proteins of resistance (R)-gene encoding conserved nucleotide binding sites (NBS) and leucine-rich repeats (LRR) can identify pathogens in cells [8,9]. ‘Gene-to-gene’ resistance resulting from the direct or indirect interaction between R-protein and oomycete effector protein is an important form of plant disease resistance, and has also become a focus of research in recent years [10,11,12].

At present, a large number of plant R-genes have been successfully cloned, and their corresponding R-proteins are very conserved in structure [10,13]. The largest class of R-genes that has been successfully cloned thus far is the NBS-LRR protein family. NBS-LRR protein-mediated resistance can effectively resist hemi-biotrophs and biotrophs, but cannot resist saprophytic pathogens that kill host cells [14,15]. NBS-LRR genes can recognize the effector proteins secreted by biotrophs, which usually cause hypersensitive response (HR) and accumulation of reactive oxygen species (ROS) in infected areas [16]. ROS directly exert antibacterial effects and activate other defense reactions [17,18]. HRs help plants to antagonize biotrophs and activate salicylic acid (SA)-dependent signaling pathways, thus activating the expression of downstream defense-related genes and the synthesis of phytoalexin [14].

The accumulation of plant hormones related to disease resistance and signal transduction is crucial in the process of plant disease resistance. At present, the SA and jasmonic acid (JA) signaling pathways as well as the ethylene (ET) and abscisic acid (ABA) signaling pathways are the most studied and best known [19,20,21]. The SA signaling pathway usually mediates plant resistance to biotrophic pathogens and the ET and JA pathways mainly mediate plant resistance to necrotrophic pathogens [22]. Knockout of *PAD4* or *EDS1*, the key genes in the SA signaling pathway, could reduce the resistance of *Arabidopsis thaliana* to *Peronospora parasitica* [23,24]. *Arabidopsis* mutants, which knockout the key gene *COI1* in the JA signaling pathway, were more susceptible to *Botrytis cinerea* [25]. In general, the SA and JA signaling pathways inhibit each other, but some genes are induced by SA and JA at the same time. In short, different signaling pathways do not work alone, but interact to form a complex signal network [26,27].

Regulation of gene expression in plants is a multi-level process affected by different factors, in which regulation at the transcriptional level is a very important key. The cis-acting elements on the promoter can regulate transcriptional initiation efficiency and gene expression because of their specific binding to transcription factors [28]. Therefore, it is of great significance to study the structure, function, and action mode of related promoters to guide transgenic breeding and improve plant traits. At present, studies on promoters responding to biological stress and abiotic stress have been reported. For example, in a study of the cloning and function of the kiwifruit *AsA*-related synthase gene promoter, it was proven that light obviously induces the activity of this kiwifruit gene promoter [29]. The core region of the *VpRPW8* promoter responsive to *Plasmopara viticola* was found by a truncation experiment [30].

Most R-genes reported can only recognize a single variant, or only a few variants, of certain pathogen species. Therefore, the development of molecular breeding may be greatly limited. Some exceptions have been reported. Over-expression of NBS-LRR genes such as *Pto* can endow plants with broad-spectrum resistance by activating the SA signaling pathway and increasing the expression of defense-related genes [31,32]. In *Arabidopsis*, overexpression of the *RPP1A* is effective against *Hyaloperonospora parasitica* and *Pseudomonas syringae* [33], and *RLM3,* encoding a TIR-NB-LRR protein, confers disease resistance to one fungal pathogen and several necrotrophic fungi [34]. Nonetheless, studies on R-genes with broad-spectrum resistance function are still scarce [34].

In our previous study, a TIR-NBS-LRR gene, *VaRGA1* cloned from *Vitis amurensis* “Shuanghong”, was induced by *Plasmopara viticola* and could improve the resistance of tobacco to *Phytophthora capsici*. [35]. These results suggested that *VaRGA1* may confer enhanced resistance to biotrophs. In this study, we found that *VaRGA1* in “Shuanghong” was induced by SA, but inhibited by JA. To further study the resistance mechanism of the gene *VaRGA1*, we transferred this gene into *Arabidopsis thaliana* plants and then treated them with biotrophic *Hpa*, hemi-biotrophic *Pst*DC3000, and the necrotrophic *B. cinerea* to determine whether the defense responses were associated with SA- and/or JA-dependent signaling pathways. Additionally, a promoter fragment of *VaRGA1* was cloned and analyzed to further elucidate the mechanism of induction of the gene at the transcriptional level. Further study on the disease resistance and signal regulation of *VaRGA1* will provide a theoretical basis for the identification of R-genes with broad-spectrum resistance function.

## 2. Results

### 2.1. VaRGA1 Expression Is Affected by SA and JA in Grape 

We have previously confirmed that *VaRGA1* was induced by *P. viticola* in *V. amurensis* “Shuanghong” [35]. In order to explore whether the expression of *VaRGA1* was affected by exogenous hormones, *VaRGA1* expression levels were detected at different time points after spraying SA and JA. Results showed that *VaRGA1* expression was strongly induced by SA at 1, 3, and 6 hours post inoculation (hpi), and it slowly reached its peak 12 h after treatment, then decreased rapidly at 24 and 48 hpi. The *VaRGA1* expression level was about 12 times higher than the control mock inoculation. In contrast, *VaRGA1* expression was suppressed by JA, decreasing to its lowest point 6 h after treatment before recovering. The expression level was less than half the mock at its lowest point, but after 48 hpi it was still lower than that of the mock (Figure 1). 

### 2.2. The Response of VaRGA1 in Transgenic Arabidopsis Lines to Different Pathogens 

In order to study the role of *VaRGA1* in the process of disease resistance, *VaRGA1* expression patterns in transgenic plants inoculated with *Hpa*, *Pst*DC3000, and *B. cinerea* were determined by quantitative real-time PCR (qRT-PCR). Results showed that gene expression was induced by *Hpa* (Figure 2A) and *Pst*DC3000 (Figure 2B), reached a peak 48 h post-inoculation (hpi), and then decreased gradually. Overall, the expression of *VaRGA1* was affected more by PstDC3000, and the expression level of *VaRGA1* was higher than that of the *Hpa*; furthermore, the amount of expression was about thrice that of the *Hpa* at 24 hpi. However, the gene expression level was inhibited by *B. cinerea* (Figure 2C), the expression level decreased gradually after infection, reached the lowest level of 48 hpi, and then increased gradually. At 96 hpi, the expression levels of this gene in L2 and L3 were both significantly lower than that before infection, and the difference was not significant for L1. These results suggest that *VaRGA1* may play different roles in the process of plant resistance to different pathogens.

### 2.3. Expression of VaRGA1 in Arabidopsis Improves Resistance to Hpa 

To investigate the role of *VaRGA1* in the process of resistance *to Hpa*, we counted the phenotype, spore number per gram, and histochemical staining of different lines. The transgenic plants showed fewer signs of necrotic leaves and spore growth seven days post-inoculation (dpi) (Figure 3A) and had lower spore numbers per gram of 5 dpi (Figure 3B) than the control (Col-0). The mutant plants showed more signs of necrotic leaves and spore growth at 7 dpi, and had significantly higher spore numbers per gram at 5 dpi than the control. Diaminobenzidine (DAB), trypan blue, and Nitro blue tetrazolium (NBT) staining were effective methods to evaluate the degree of disease resistance of the plants, and were used to detect H_2_O_2_ accumulation, cell death, and superoxide anion (O_2_^−^) accumulation, respectively. The depth of staining was positively correlated with the disease resistance of plants. Staining results showed that transgenic plants had more H_2_O_2_ and O_2_^−^ accumulation than the control, while the mutants had less H_2_O_2_ and O_2_^–^ accumulation than the control (Figure 3C). Mock treatments of histochemical staining following inoculation with *Hpa* are shown in Appendix A. These results suggest that expression of *VaRGA1* can enhance the resistance of *A. thaliana* to *Hpa*. 

### 2.4. Assessment of Defense-Related Gene Expression After Inoculating Hpa 

SA-dependent defenses play an active role in plant resistance to *Hpa*, an *Arabidopsis* biotroph. The expression of *AtNPR1* (Figure 4A) and *AtEDS1* (Figure 4B), the key genes of SA signaling pathway in different lines, was measured at different time points after inoculation. The relative expression level of both genes was significantly higher in transgenic lines than in the control (Col-0) before inoculation, but was significantly lower in *varga1*. The expression of *AtNPR1* (Figure 4C) and *AtEDS1* (Figure 4D) in the transgenic lines reached peaks at 48 and 24 hpi, respectively, and then decreased gradually. The expression of the two genes decreased significantly at 96 hpi, close to the level before the inoculation. Col-0 and *varga1* also showed similar trends, but had significantly lower expression levels than transgenic lines. Therefore, the order of expression of both genes was L1 > L2 >L3 > Col-0 > *varga1* (mutants). JA-dependent defense is another signaling pathway involved in plant disease resistance. Results showed that the key genes in this pathway, *AtPR3* and *LOX3,* were slightly induced by *Hpa*, their expression reaching a peak at 48 hpi, but the degree of induction of transgenic lines was significantly lower than that of the control (Col-0), while that of the mutants was significantly higher than that of the control. The expression of *AtNPR1* and *AtEDS1* of all lines reached the highest level at 48 hpi and decreased close to the level before the inoculation at 48 hpi. 

### 2.5. Expression of VaRGA1 in Arabidopsis Improves Resistance to PstDC3000

To explore whether *VaRGA1* plays an important role in the anti-*Pst*DC3000 mechanism of *A. thaliana*, the leaf phenotype, spore number, and histochemical staining of 5-week-old seedlings after inoculation were assessed. Only a small amount of mottled yellowing was observed in the leaves of transgenic lines, compared with the large area of yellowing in the leaves of Col-0 and the yellowing and even wilting in the leaves of *atrga1* at 5 dpi (Figure 5A). On the second day after inoculation, the spore number of the transgenic lines was significantly lower than that of Col-0 and *atrga1* (Figure 5B). The results of trypan blue staining at 3 dpi showed that the leaves of the transgenic lines were darker blue when compared to those of Col-0 and *atrga1*, indicating that the local leaf necrosis area of the transgenic lines in the early stage of infection was larger than that of Col-0 and *atrga1*. The results of DAB staining showed that the yellowing area of the leaves was larger and the accumulation of reactive oxygen species (ROS) was higher in the transgenic lines than in Col-0 and *atrga1*. Aniline blue staining showed that the transgenic lines had more callose accumulation than did Col-0 and *atrga1* (Figure 5C). Mock treatments of histochemical staining following inoculation with *PstDC3000* were shown in Appendix A. These results showed that the transgenic lines were more resistant to *Pst*DC3000 than Col-0 and *atrga1*, with the following order of disease resistance: L1 > L2 >L3> Col-0 > *atrga1*.

### 2.6. Assessment of the Expression of Defense-Related Genes after Inoculating PstDC3000

The expression levels of key genes *AtNPR1* (Figure 6A) and *AtEDS1* (Figure 6B) in the SA signaling pathway were detected at different time points after the inoculation of *Pst*DC3000 to explore the signaling pathway involved in the resistance of transgenic *A. thaliana*. The expression of *AtNPR1* and *AtEDS1* in the transgenic lines increased rapidly after infection, and was significantly higher than that in the control (Col-0) and mutant (*atrga1*), reached a peak at 48 and 24 hpi, respectively, and then decreased gradually. The expression of *AtNPR1* decreased slowly, and its expression level was 5–10 times higher than that before infection at 96 hpi. The expression of *AtEDS1* was at a lower level at 72 hpi and was close to the level before the inoculation at 96 hpi. Although the expression of the key genes *AtPR3* (Figure 6C) and *LOX3* (Figure 6D) in the JA signaling pathway of all lines increased after inoculation, the expression level in the transgenic lines was significantly lower than that in Col-0, especially at 48 and 72 hpi, but the expression level was significantly higher in *atrga1* than in Col-0. These results showed that the SA and JA signaling pathways both acted in the anti-*Pst*DC3000 response of *A. thaliana*, but the intensity of the former was greater. 

### 2.7. Expression of VaRGA1 in Arabidopsis Decreases Resistance to B. cinerea

The resistance of transgenic lines of *VaRGA1* to *B. cinerea* was judged by assessing the area of leaf necrosis (Figure 7A) and diameter of lesions (Figure 7B) at 3 dpi. Not only was the disease spot diameter of the transgenic lines significantly larger than that of Col-0, but the degree of leaf infection was also more obvious, resulting in the attachment of large areas of *B. cinerea*. However, mutants were not as sensitive as Col-0. *B. cinerea* is a necrotrophic pathogen, and the accumulation of dead cells and ROS promote the infection. Histochemical staining (Figure 7C) showed that cell death and the accumulation of H_2_O_2_ and O_2_^−^ in transgenic lines was significantly higher than in Col-0. Mock treatments of histochemical staining following inoculation with *B. cinerea* are shown in Appendix A. All experimental results showed that overexpression of *VaRGA1* could decrease the resistance of *A. thaliana* to *B. cinerea*. 

### 2.8. Assessment of the Expression of Defense-Related Genes After Inoculating B. cinerea

To fight against *B. cinerea*, the JA signaling pathway of plants is activated to initiate a series of downstream defense reactions. The role of *VaRGA1* in the anti-*B. cinerea* response of *A. thaliana* was judged by detecting the expression of the key genes of the SA and JA signaling pathways after inoculation. The expression levels of *AtNPR1* (Figure 8A), a key gene in the SA signaling pathway, was reduced to its lowest level at 48 hpi, followed by a rapid increase at 72 hpi and a decrease at 96 hpi. The expression of the key genes *AtEDS1* (Figure 8B) in the SA signaling pathway of all lines peaked at 24 hpi and then gradually decreased. The expression of *AtNPR1* and *AtEDS1* was significantly higher in the transgenic lines than in Col-0 and *atrga1* during the whole process. The expression levels of the two key genes, *AtPR3* (Figure 8C) and *LOX3* (Figure 8D), in the JA signaling pathway significantly increased after inoculation in the transgenic lines, but remained significantly lower in Col-0 and *atrga1*. The expression level of *AtPR3* and *LOX3* continued to increase in non-transgenic lines and reached maximums at 96 hpi. For the transgenic lines, the expression levels of *AtPR3* and *AtLOX3* in different lines were different, but on the whole, expression levels of *AtPR3* and *AtLOX3* had increased slowly and reached the maximum at 72 hpi, and then a certain degree of decrease was observed at 96 hpi.

## 3. Discussion

The results of our study showed that *VaRGA1* expression in “Shuanghong” induced the SA pathway, inhibited the JA pathway, and additionally, *VaRGA1* expression in the transgenic *A. thaliana* was induced by *Hpa* and *Pst*DC3000 and inhibited by *B. cinerea*. These results suggest that *VaRGA1* can enhance disease resistance and, in particular, resistance to biotrophic pathogens through the SA signaling pathway. The largest R-protein type, the NBS-LRR protein family in plants, has been recognized to stimulate immune responses caused by effectors [8,36,37]. The NBS-LRR-type R-genes are expressed at low levels before the pathogen attacks, but once the pathogen invades plant tissues or hormone levels change, they can be rapidly induced to an appropriate level to play a role in disease resistance [38,39,40]. For example, the low expression level of one coiled-coil (CC)-NBD-LRR-like R-gene cloned from riparian grape (*V. riparia*) under normal growth was significantly improved after inoculation with *P. viticola* [41]. The expression of *Arabidopsis RPP8* was rapidly increased following infection by *Hyaloperonospora arabidopsidis* or spraying with SA hormone to respond to such biological or non-biological stress factors [42]. Our previous study showed that the expression of *VaRGA1* in *V. amurensis* “Shuanghong” was first rapidly increased and then decreased slowly after inoculation with *Hpa* [35]. 

The regulation of transcription level is the most important mode of gene expression, but the regulation of post-transcriptional level also plays an important role in the process of gene expression. Some nuclear proteins such as STA1 are crucial for pre-mRNA splicing and the turnover of unstable transcripts in plant responses to different stresses [43]. Whether eukaryotes can use mature mRNA molecules to translate proteins for growth and development for a long time is closely related to the stability of mRNA and the release of shielding state. Under biotic and abiotic stresses, specific smRNAs were induced to regulate the mRNA stability by intersecting with most of the pathways [44]. The stability of SOS1 mRNA was greatly improved under the treatment of H_2_O_2_, salt and other ionic and dehydration [45]. The level of mRNA in *VaRGA1* in this study was affected by different pathogenic bacteria, possibly because of the change in mRNA stability under stress conditions. Since many of the mRNA in the eukaryotes are extremely stable, the transcription of the strong promoter and the regulation of the specific smRNAs may also increase the level of mRNA of this gene under certain stress conditions.

*B. cinerea*, a common necrotrophic pathogen, can cause diseases in many plants by secreting virulence factors and cell wall degradation enzymes [40,46]. Few R-genes with resistance to *B. cinerea* have been reported because effector-triggered immunity induced by R-gene can activate the plant HR reaction in the early stage of pathogen infection, which is helpful to the infection and growth of necrotrophic pathogens [47,48]. This study further supported the above conclusions, showing that transgenic *A. thaliana* overexpressing *VaRGA1* had lower resistance to *B. cinerea*, larger lesion area, and more spores than the other lines. However, the *atrga1* mutant could improve the resistance of *A. thaliana* to *B. cinerea*. *B. cinerea* is tolerant to oxidative burst, and can produce ROS to promote its infection. ROS, signaling molecules that promote cell death, are an indicator of successful infection by *B. cinerea* [49]. Histochemical staining showed that the transgenic lines had more cell death and reactive oxygen accumulation than other lines, which helped the spread of *B. cinerea*. Moreover, the R-gene *RPW8*, which regulates resistance to powdery mildew in *A. thaliana*, can decrease the resistance of *A. thaliana* to *B. cinerea* [50], which is consistent with our results.

NBS-LRR genes can recognize the effectors secreted by biotrophic pathogens to mediate resistance [36]. They usually cause hypersensitivity in the place of infection, followed by ROS production, accumulation of plant hormones related to disease resistance, expression of defense-related genes, and phytophane synthesis [51,52]. The production of ROS is a necessary condition for cell death, which can restrict biotrophic pathogens from obtaining nutrients from plants [53,54]. Therefore, continuous expression of CC-NBS-LRR gene *ADR1* in *A. thaliana* enhanced the resistance to biotrophic pathogens [55]. In our study, overexpression of *VaRGA1* could enhance the resistance of *A. thaliana* to *Hpa* and *Pst*DC3000, and significantly reduce the number of spores of these two bacteria. The results of histochemical staining showed that transgenic lines had more cell death and ROS accumulation than other lines, which helped limit the growth of biotrophic pathogens. The mutant *atrga1* was more susceptible than Col-0 to *Hpa* and *Pst*DC3000, which further proved the resistance of *VaRGA1* to these two bacteria. However, as previously proposed, the degree of cell death and accumulation of ROS in *VaRGA1* transgenic lines were significantly higher than those in Col-0, which led to the decrease of resistance to *B. cinerea*.

Mock is one of the most basic principles of experimental design, which is essential for obtaining reliable experimental results. The leaves of the mock in the *Hpa* inoculation experiment were close to transparency after staining. However, as the leaves were too small, they needed to be observed under the microscope. It was difficult to take the actual color because of the influence of microscope light. The color of stained mock leaves was uniform and lighter under the microscope, while that treated with *Hpa* was darker, with mottled spots. Therefore, mock treatment was effective and could reduce the influence of various uncertain factors in the experiment. The leaves treated with *PstDC3000* and *B. cinerea* were larger and could be photographed in normal light after staining, so the mock could reduce the experimental error more intuitively. For the mock in the *B. cinerea* inoculation experiment, clear callose was observed by NBT staining, which might be due to the fact that the leaves to be treated need to be placed in vitro, causing their own defense and response. Even so, the callose content of the mock was significantly lower than that with other treatments, so the experimental results were reliable. 

As a result of the lack of mobile defense cells and acquired immune systems, plants can only rely on the innate immunity of each cell and the systemic signals sent from the place of infection to resist pathogens [56,57,58]. The accumulation of plant hormones and signal transduction related to disease resistance are important links in the mechanism of plant disease resistance. At present, the known plant disease resistance signaling pathways mainly include the SA and JA signaling pathways, and different signaling pathways intersect into a complex signal network [59]. These two signaling pathways are usually suppressed by each other, but there have also been reports of synergy [60,61]. In this study, the expression of *AtNPR1* and *AtEDS1* (the key genes of the SA signaling pathway) significantly increased after inoculation with *Hpa*, *Pst*DC3000, and *B. cinerea*. The expression of *AtPR3* and *AtLOX3*, the key genes of the JA pathway, increased slightly, but was significantly lower than that of Col-0. In other words, the JA signaling pathway was suppressed to a certain extent. It can be inferred that *VaRGA1* can activate the SA signaling pathway during pathogen infection, which can increase the resistance of *A. thaliana* to *Hpa* and *Pst*DC3000, but decrease the resistance to *B. cinerea*. Previous studies have also shown that SA and its signaling pathway genes are induced by some NBS-LRR genes, which are an important component of the plant disease resistance system [62,63]. 

A promoter is an important element in the regulation of the transcription process, which largely determines the expression level of downstream genes [64,65]. HSC70 is a highly conserved molecular chaperone, and its promoter activity increased at high temperature, which explains the rapid increase of the expression of the HSC70 gene at high temperature [66]. The *STS* promoter in *V. pseudoreticulata* could respond to various biological and non-biological stress [64]. In our study, a 1104 bp promoter fragment of *VaRGA1* was cloned and analyzed. In addition to common elements such as CAAT-box and TATA-box, some hormone-induced elements and stress elements were also predicted (Appendix A). TCA-element was involved in SA responsiveness, and MBS was involved in drought-inducibility. These elements are likely to promote the expression of *VaRGA1* under certain conditions.

Currently, only a few R-genes with broad-spectrum resistance function have been reported, which greatly limits the development of molecular breeding. Fortunately, some studies have found that overexpression of NBS-LRR genes such as *NLS1* and *WRR4* can activate the SA signaling pathway and increase the expression of defense-related genes allowing broad-spectrum resistance [67,68]. Similarly, we have demonstrated that *VaRGA1* could improve the resistance of transgenic *A. thaliana* to *Hpa* and *Pst*DC3000 by activating the SA signaling pathway, but decrease the resistance to *B. cinerea*. In summary, our study preliminarily confirmed the disease resistance function of *VaRGA1* and its signal regulation pathway, and made some contributions to the identification of R-genes with broad-spectrum resistance function. However, its detailed mechanism needs to be further studied. Future research will focus on finding the core region of the upstream promoter of *VaRGA1* and the protein that interacts with the gene.

## 4. Materials and Methods 

### 4.1. Plant Materials and Pathogens

The annual *V. amurensis* “Shuanghong”, wildtype *A. thaliana* (Col-0), and *pad4* mutant were kindly provided by our laboratory. The mutants of *A. thaliana* gene *AT5G36930.* (*atrga1*), which are homologous to *VaRGA1*, were purchased from NASC [69]. *V. amurensis* “Shuanghong” was planted in a flowerpot containing a 1:1 mixture of nutritious soil and vermiculite, then grown in a greenhouse at a temperature of 25 °C and a 16-h light/8-h dark cycle. The plants were watered every week and fertilizer was applied every other month. *Arabidopsis* were grown in a chamber at 25 °C and a 16-h light/8-h dark photoperiod. *Hpa* (*At*) was cultured on 10-day-old *pad4* mutants. *Pst*DC3000 was maintained on lysogeny broth (LB) medium containing 50 mg·L^−1^ rifampicin at 30 °C. *B. cinerea* was maintained on a potato dextrose agar (PDA) medium containing 100 mg·L^−1^ kanamycin at 23 °C in the dark. 

### 4.2. Hormone Treatments in Grape 

Solutions of 100 mM SA (Tokyo, Japan) and 50 mM methyl jasmonate (MeJA, Haarlem, The Netherlands) were sprayed on grape leaves with the same growth conditions, and aseptic ultra-pure water was sprayed as the mock [70]. The expression level of *VaRGA1* was determined at 0, 1, 3, 6, 12, 24, and 48 h post-treatment.

### 4.3. Generation of Transgenic Plants

The coding sequence of *VaRGA1* was amplified by PCR using the vector named *pBI121-VaRGA1-GFP* from our previous work [35]. The specific homologous recombination primers were pHB*-VaRGA1-GFP-F* (5′-agcttggatccagaactagtATGTCAATCGGCATGGATC-3′) and pHB*-VaRGA1-GFP-R* (5′-cccttgctcaccatactagtTGCAAAAGAGAGCAAAGTTC-3′). The lowercase letters in the above primers represent the nucleotides with restriction sites, and the uppercase letters represent the nucleotides complementary to the *VaRGA1* gene. The PCR product was then cloned downstream of the cauliflower mosaic virus (CaMV) 35S promoter of pHB-35S:*EGFP*, a plant over-expression vector. Subsequently, the recombinant plasmid named pHB*-VaRGA1-GFP* was introduced into *Agrobacterium* (strain GV3101) that were used to transform *A. thaliana* (Col-0), according to an existing protocol [71]. Seventy transgenic lines were obtained, and three lines (L1, L2, and L3) with the highest expression induced by *Hpa* were chosen for selecting T4 generation homozygous plants for subsequent experiments. RT-PCR detection and subcellular localization of the transgenic lines are shown in Appendix A.

### 4.4. Inoculation of Pathogens

The downy mildew (causal agent *Hpa*) treatment was conducted following a previously described method [42]. The seeds of transgenic, mutant, and wild type *A. thaliana* were grown evenly in fertile soil at 25 seeds per pot. *Arabidopsis* downy mildew solution was evenly sprayed on the leaves and cultured at 90% humidity for seven days. Histochemical detection and spore number calculation were carried out at five days post inoculation (dpi). Leaf necrosis area and spore growth were observed at 7 dpi. 

*Pst*DC3000 was activated by fresh LB liquid medium (28 °C, 200 rpm) [72]. The optical density at 600 nm (OD600) of the secondary activation was 0.6. After bacterial cell collection (5000 rpm, 5 min), cells were suspended in aseptic 10 mm MgCl_2_ solution (0.05% Silwet-77) and OD600 adjusted to 0.004. Four-week-old *A. thaliana* plants were selected and the backs of leaves were injected with *Pst*DC3000 using a 1-mL aseptic syringe. After inoculating all leaves, the plants were placed in a black tray, covered with white transparent plastic, and sprayed with water to moisturize properly [72]. Spore number statistics were assessed at 2 dpi, and histochemical detection was carried out 3 dpi. Area of necrosis and yellowing of the leaves were assessed at 5 dpi.

The *B. cinerea* inoculation method followed a previously described protocol [73]. First, *B. cinerea* was cultured on PDA culture medium at 23 °C. After about 21 days, the mycelium and the spore were suspended in the inoculation liquid of 4% maltose and 1% peptone. The residual body of the hypha was removed by filtration. Inoculant of the mycotic spore suspension was prepared by counting and adjusting the concentration of the mycotic spores to 2−106 mL^−1^ with a blood cell counting plate. Four-week-old leaves of *A. thaliana* were cut to inoculate 10 μL spore suspension on the near-axis surface. Water mist was sprayed in the tray and was moisture-maintained by a preservative film. Leaves were cultured at 23 °C. Histochemical detection, spore number statistics, and the disease state of the leaf blade were conducted 3 dpi. 

To detect H_2_O_2_ accumulation, plant materials were put in 1 mg·mL^−1^ DAB for overnight dyeing in the dark. Then, 70% ethanol was used for decolorizing. Cell death was assessed by trypan blue staining [74]. The materials were placed in boiled trypan blue solution (a 1:1:1:1:1 ratio of ddH_2_O, trypan blue, phenol, glycerol, and lactic acid) for 5 min and decolorized in 2.5 g·mL^−1^ chloral hydrate for 1–2 days. Superoxide anion (O_2_^−^) accumulation can be detected by NBT staining [75]. Additionally, 6 mM NBT was dissolved in Hepes buffer solution (pH 7.5) and the leaves of *Arabidopsis* were dyed in NBT dyeing solution for 2 h. Then, leaves were decolorized in a 95% ethanol water bath at 50 °C for 1 hour. 

### 4.5. Expression Analysis of Related Genes by Quantitative Real-Time PCR 

The total RNA of plant materials was extracted using a Plant RNA Kit (R6827-02; Omega Bio-tek, GA, USA). RNA purity and concentration were detected by 1% agarose gel electrophoresis and nanodrop2000, respectively. Reverse transcription of the extracted RNA was completed with TransScript (AT311-03; TaKaRa Biotechnology, Dalian, China), and the obtained cDNA was diluted six times. Quantitative real-time PCR (qRT-PCR) was conducted using a previously described method [35]. The primers used for qRT-PCR are listed in Appendix A. Leaf samples were collected at different time points for subsequent gene expression analyses. For the detection of *VaRGA1* expression induced by SA and JA in grape, leaf samples were collected at 0, 1, 3, 6, 12, 24, and 48 hpi. To detect the expression of *VaRGA1* and defense-related genes in transgenic Arabidopsis lines after inoculating different pathogens, leaf samples were collected at 0, 24, 48, 72, and 96 hpi.

### 4.6. Plant DNA Extraction and VaRGA1 Promoter Cloning

Extraction of the genomic DNA of *V. amurensis* “Shuanghong” was conducted using the hexadecyl trimethyl ammonium Bromide (CTAB) method. Primers *(PVa-F* and *PVa-R;* listed in Appendix A) were designed according to the previously cloned *VaRGA1* to obtain 1242 bp PCR products including specific upstream sequences and coding sequences (CDS). PCR amplification and vector construction were conducted following a previously described method [76]. The obtained sequences were analyzed by PlantCARE [77]. 

### 4.7. Statistical Analysis

Three biological replicates and three technical replicates were conducted for each experiment. Data analysis and plotting were conducted in Microsoft Excel (Microsoft Corporation, Redmond, WA, USA) and SigmaPlot 14.0 (Systat, Inc., Point Richmond, CA, USA). Significant differences were detected by SPSS Statistics 17.0 (IBM China Company Ltd, Beijing, China). (Student’s *t* test, *p* < 0.01). 

## Figures and Tables

**Figure 1 ijms-21-00193-f001:**
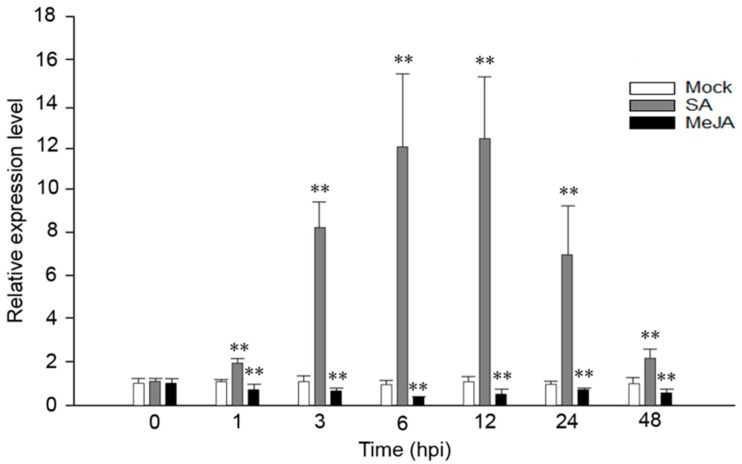
Expression levels of *VaRGA1* in *V. amurensis* “Shuanghong” following salicylic acid (SA) and methyl jasmonate (MeJA) treatments. The expression level of *VaRGA1* in “Shuanghong” under non-stressed conditions was defined as 1.0. Data represent mean values ± SD from three independent experiments. Asterisks show statistically significant difference (** *p* < 0.01, Student’s *t* test).

**Figure 2 ijms-21-00193-f002:**
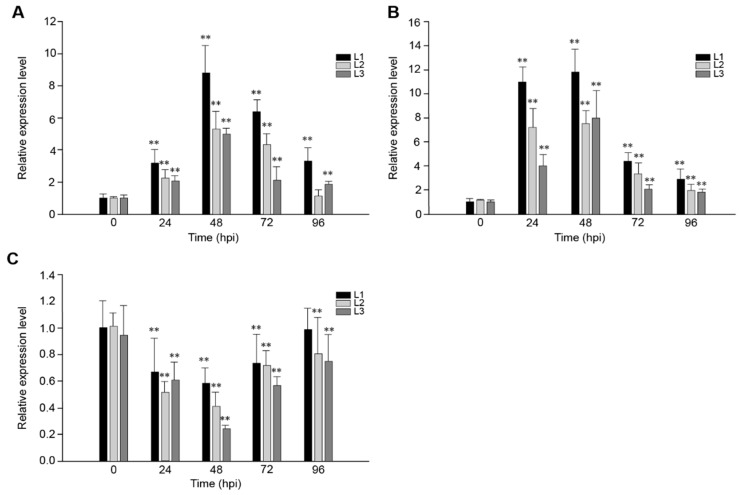
*VaRGA1* expression level in transgenic *Arabidopsis* following (**A**) *Hpa* infection, (**B**) *Pst*DC3000 infection, and (**C**) *B. cinerea* infection. *VaRGA1* expression level in transgenic *Arabidopsis* line 1 (L1) under non-stressed conditions was defined as 1.0. Data represent mean values ± SD from three independent experiments. Asterisks show statistically significant differences (** *p* < 0.01, Student’s *t* test).

**Figure 3 ijms-21-00193-f003:**
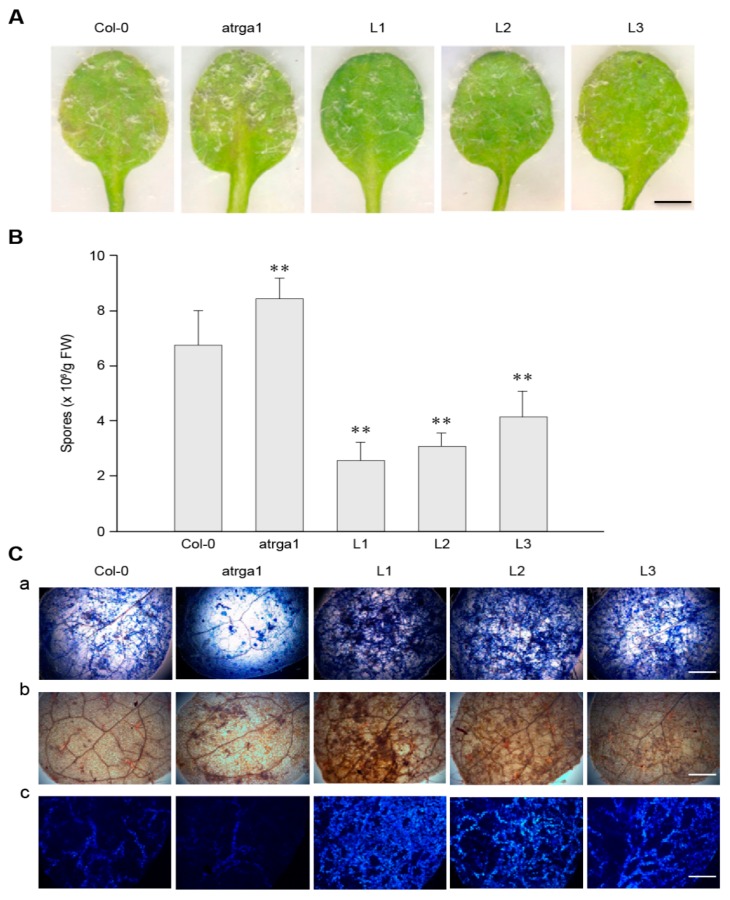
*VaRGA1* expression in *Arabidopsis* improves resistance to *Hpa*. Transgenic plants (L1, L2, L3), mutants (*atrga1*), and non-transgenic controls (Col-0) were inoculated with *Hpa* for the following experiments. (**A**) The symptom of plants infected with *Hpa* seven days post-inoculation (dpi). Scale bars = 1 mm. (**B**) Biological statistics on the number of spores per gram were assessed at 5 dpi. Data represent mean values ± SD from three independent experiments. Asterisks show a statistically significant difference (** *p* < 0.01, Student’s *t* test). (**C**) Histochemical staining was performed to detect cell death, H_2_O_2_ accumulation, and O_2_^−^ accumulation at 5 dpi with (a) trypan blue, (b) diaminobenzidine (DAB), and (c) Nitro blue tetrazolium (NBT) staining, respectively. Three independent experiments were conducted with 10 leaves in each. Scale bars = 0.5 mm.

**Figure 4 ijms-21-00193-f004:**
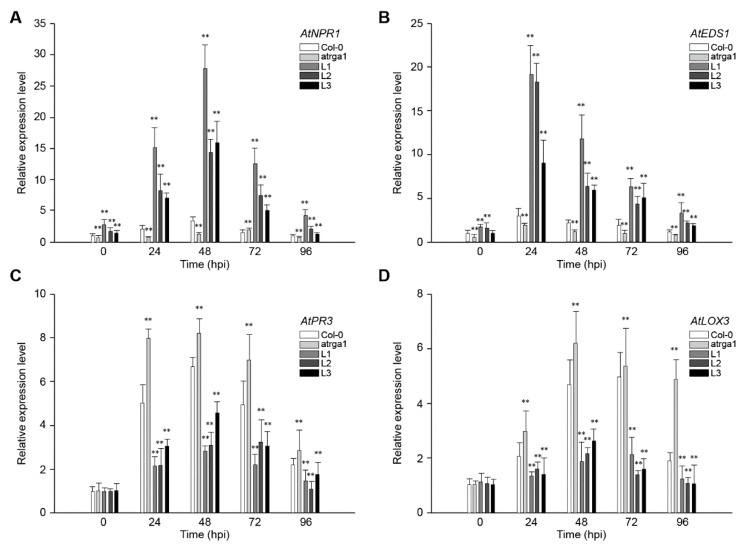
Assessment of the expression of defense-related genes at different time points after inoculating *Hpa.* Relative expression levels of (**A**) *AtNPR1,* (**B**) *AtEDS1,* (**C**) *AtPR3,* and (**D**) *LOX3* were detected via qRT-PCR. Data represent mean values ± SD from three independent experiments. Asterisks show statistically significant differences (** *p* < 0.01, Student’s *t* test).

**Figure 5 ijms-21-00193-f005:**
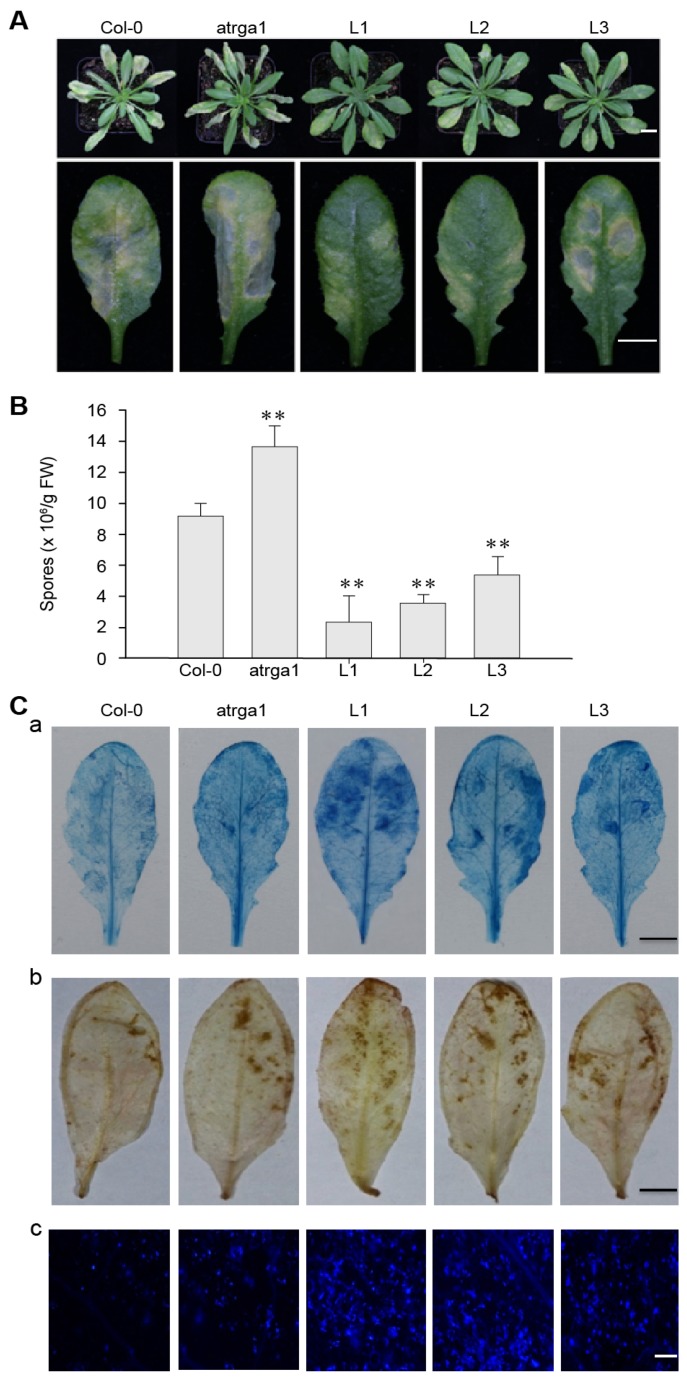
*VaRGA1* expression in *Arabidopsis* improves resistance to *Pst*DC3000. Transgenic plants (L1, L2, L3), mutants (*atrga1*), and non-transgenic controls (Col-0) were inoculated with *Pst*DC3000 for the following experiments. (**A**) The symptom of plants infected with *Pst*DC3000 5 dpi. The first scale bars = 20 mm. The second scale bars = 10 mm (**B**) Biological statistics on the number of spores per gram was carried out 2 dpi. Data represent mean values ± SD from three independent experiments. Asterisks show statistically significant difference (** *p* < 0.01, Student’s *t* test). (**C**) Histochemical staining was carried out for the detection of cell death, H_2_O_2_ accumulation, and superoxide anions (O2^−^) accumulation 3 dpi for (a) trypan blue, (b) diaminobenzidine (DAB), and (c) nitro blue tetrazolium (NBT) staining, respectively. Three independent experiments were conducted with 10 leaves per experiment. The black scale bars = 10 mm. The white scale bars = 0.5 mm.

**Figure 6 ijms-21-00193-f006:**
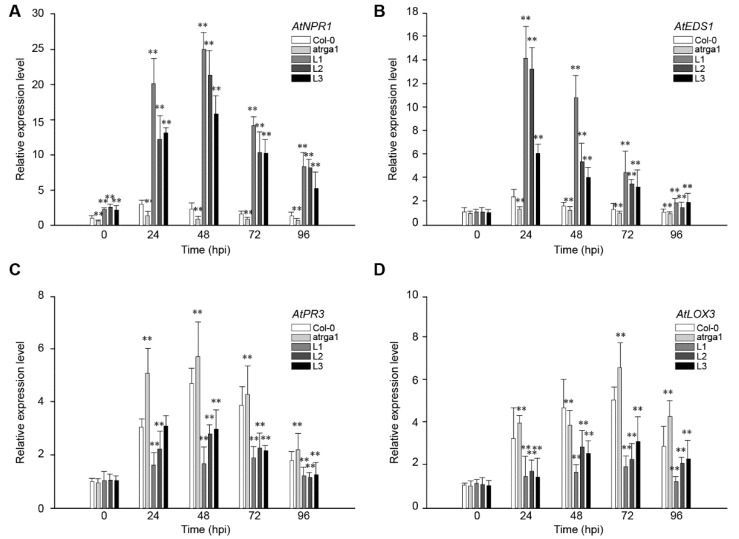
Assessment of the expression of defense-related genes at different time points after inoculating *PstDC3000.* Relative expression levels of (**A**) *AtNPR1;* (**B**) *AtEDS1;* (**C**) *AtPR3*, and (**D**) *LOX3* were detected via qRT-PCR. Data represent mean values ± SD from three independent experiments. Asterisks show the statistically significant difference (** *p* < 0.01, Student’s *t* test).

**Figure 7 ijms-21-00193-f007:**
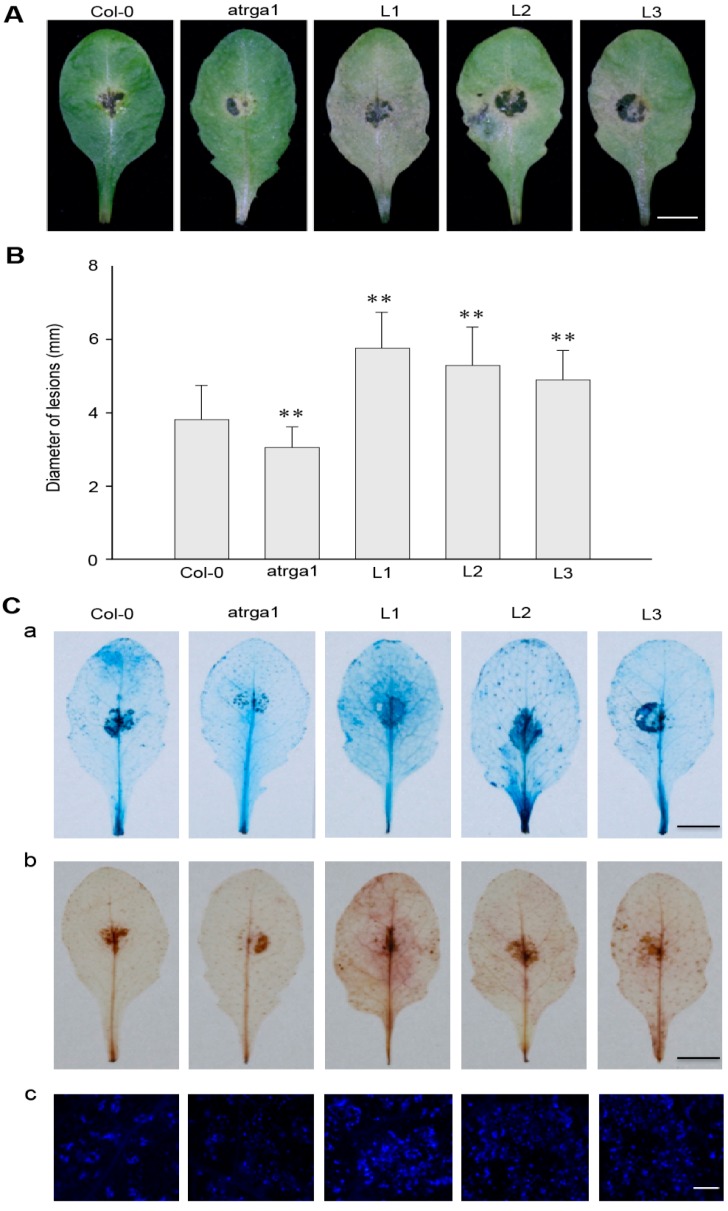
*VaRGA1* expression in *Arabidopsis* decreases resistance to *B. cinerea*. Transgenic plants (L1, L2, L3), mutants (*atrga1*), and non-transgenic controls (Col-0) were inoculated with *B. cinerea* for the following experiments. (**A**) Symptoms of plants infected with *B. cinerea* 3 dpi. Scale bars = 10 mm (**B**) Biological statistics on lesion diameters were assessed at 3 dpi. Data represent mean values ± SD from three independent experiments. Asterisks show statistically significant differences (** *p* < 0.01, Student’s *t* test). (**C**) Histochemical staining was performed to detect cell death, H_2_O_2_ accumulation, and superoxide anions (O_2_^−^) accumulation at 3 dpi with (a) trypan blue, (b) diaminobenzidine (DAB), and (c) Nitro blue tetrazolium (NBT), respectively. Three independent experiments were conducted with 10 leaves in each. The black scale bars = 10 mm. The white scale bars = 0.5 mm.

**Figure 8 ijms-21-00193-f008:**
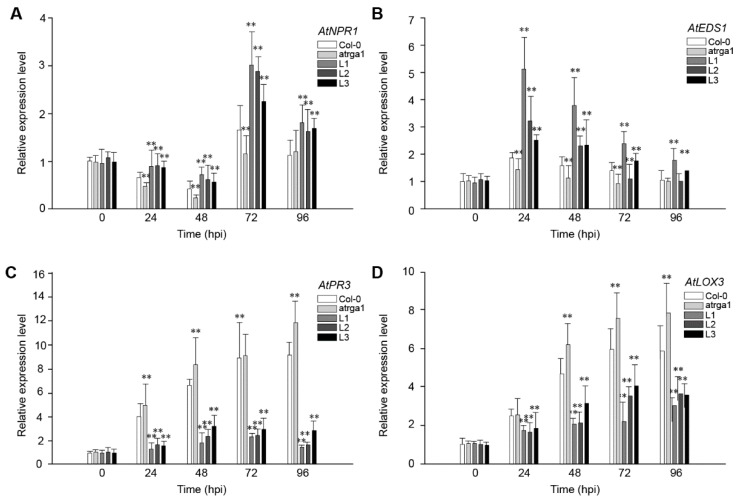
Assessment of the expression of defense-related genes at different time points after inoculating *B. cinerea.* Relative expression levels of (**A**) *AtNPR1;* (**B**) *AtEDS1*; (**C**) *AtPR3*, and (**D**) *LOX3* were detected via qRT-PCR. Data represent mean values ± SD from three independent experiments. Asterisks show the statistically significant difference (** *p* < 0.01, Student’s *t* test).

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
