# Peer review of "Ectopic Expression of Grapevine Gene VaRGA1 in Arabidopsis Improves Resistance to Downy Mildew and Pseudomonas syringae pv. tomato DC3000 But Increases Susceptibility to Botrytis cinerea"

_ijms, 2019, doi:10.3390/ijms21010193_

Round 1
Reviewer 1 Report
The topic of the paper is surely of interest if we consider the large use of pesticides for grape cultivation. The use of Arabidopsis as model species is acceptable since V.vinifera is still difficult to transform. My main criticism is about the way the paper is presented. It is extremely confusing very difficult to read and there is no clear explanation of the experimental plan. Moreover, many pieces of information about molecular characterization of transgenic lines are missing (to give an example: how many transgene copies were inserted?). Also, expression analysis is not well detailed in Results and it is necessary to go to MM to understand at which time points the material was sampled and from which part of the plant and so on.
Author Response
Comments and Suggestions for Authors
The topic of the paper is surely of interest if we consider the large use of pesticides for grape cultivation. The use of Arabidopsis as model species is acceptable since V.vinifera is still difficult to transform. My main criticism is about the way the paper is presented. It is extremely confusing very difficult to read and there is no clear explanation of the experimental plan. Moreover, many pieces of information about molecular characterization of transgenic lines are missing (to give an example: how many transgene copies were inserted?). Also, expression analysis is not well detailed in Results and it is necessary to go to MM to understand at which time points the material was sampled and from which part of the plant and so on.
Response:(1)The research purpose of each experiment and the relationship between them were described in more detail. They were shown in the last paragraph of Introduction.
(2) We added more detailed information about transgenic lines, including RT-PCR detection and subcellular localization of the gene expression protein (Supplementary Figure 1 and Supplementary Figure 2).
(3) The expression of related genes was analyzed in more detail. Details were shown in Results.
(4) The detailed information of sampling time and sampling position in the experiment was supplemented. Details were shown in Materials and Methods.
(5) Professional English editing service has been used for extensive English editing.
We tried our best to improve the manuscript and made some changes in the manuscript.. These changes will not influence the content and framework of the paper. Thank you very much for your comments and suggestions.
Reviewer 2 Report
Tian et al. examine VaRGA1’s function in disease resistance by overexpression approach using Arabidopsis. The transgenic Arabidopsis lines overexpressing VaRGA1 show improved resistance to P. parasitica and PstDC3000, while they are more susceptible to a necrotrophic pathogen B. cinerea. VaRGA1-overexpressing Arabidopsis plants tend to show higher ROS accumulation and more cell death, and induce some key genes in salicylic acid (SA) and jasmonic acid (JA) pathway upon disease infection more strongly than wild type and a mutant line supposedly defective in a VaRGA1-homologous gene (atrga1). Although this study has a potential to provide new insights in molecular function of VaRGA1 in SA/JA-mediated disease resistance in plants, the current manuscript bears several major concerns and confusing points as listed below, which should be addressed before being considered for publication in IJMS.
Specific comments
(1) In the transgenic Arabidopsis lines, how does infection with P.parasitica or PstDC3000 (Fig.2) “induce” VaRGA1 expression from a constitutively overexpressing 35S promoter? According to the method section, VaRGA1 cds was directly placed under 35S promoter. Is there any post-transcriptional regulation such as mRNA turnover for VaRGA1 transcript? Or did the cloning include a cis element conferring the inducibility?
(2) Although this study addresses new aspects of VaRGA1 function in JA-mediated defense and different types of diseases, the conclusion regarding VaRGA1 improve SA-mediated defense was already shown in the group’s previous publication (Li, X. et al. 2017) using a very similar overexpression approach using N. benthamiana. I feel the manuscript will greatly benefit from examining the VaRGA1’s function in a little more detail, taking advantage of genetic resources of Arabidopsis. For example, would VaRGA1 be able to improve disease resistance in mutants defective in SA/JA pathways? Or is it completely dependent on the hormone response pathways?
(3) As examined in N. benthamiana in their previous publication, subcellular localization of the over-expressed VaRGA1 in the transgenic Arabidopsis lines should be examined. Does VaRGA1 localize in both nucleus and cytoplasm in Arabidopsis? Does disease infection change the localization pattern? Would excluding VaRGA1 from nucleus by a nuclear export signal abolish the observed effect of VaRGA1 in the transgenic lines?
(4) The nature of “atrga1” mutant is not clear. There is no Arabidopsis gene with ID “ATG43098” (l.356). Please provide correct accession number and identification number for the “atrga1” mutant line. “homologous” should be explained as well. How about % identity and similarity to VaRGA1? Are there multiple genes with comparable similarity/identity? If so, how did the authors select one?
(5) For histochemical staining data, mock treatment for each genotype should be included (Fig. 3C, 5C, 7C) to show the basal level of H2O2/cell death/O2- in each line.
(6) At 0-hpi, AtNPR1 and AtEDS1 expression show statistical difference among genotypes in some experiments but not in the others (Fig. 4,6,8). Is this due to different condition of 0-hpi plants for each pathogen? It would be worth clarifying a condition where the key hormone regulatory genes are altered even in the absence of biotic stresses.
(7) Fig. 9 identifies potential cis-regulatory element candidates, but the search result alone without an empirical support is not a strong argument for their actual functions. In fact, VaRGA1 cds without any of these elements still show some induction upon infection (Fig.2). Authors can validate the importance (or lack thereof) of each candidate element by a reporter gene assay. Without that, I suggest making Fig.9 a supplemental figure.
(8) l.98: VaRGA1 expression is suppressed, not induced, by JA in Fig. 1.
(9) l.89: [35] should be [34]? The reference [34] did not seem to mention about P. capsici.
Author Response
Comments and Suggestions for Authors
Tian et al. examine VaRGA1’s function in disease resistance by overexpression approach using Arabidopsis. The transgenic Arabidopsis lines overexpressing VaRGA1 show improved resistance to P. parasitica and PstDC3000, while they are more susceptible to a necrotrophic pathogen B. cinerea. VaRGA1-overexpressing Arabidopsis plants tend to show higher ROS accumulation and more cell death, and induce some key genes in salicylic acid (SA) and jasmonic acid (JA) pathway upon disease infection more strongly than wild type and a mutant line supposedly defective in a VaRGA1-homologous gene (atrga1). Although this study has a potential to provide new insights in molecular function of VaRGA1 in SA/JA-mediated disease resistance in plants, the current manuscript bears several major concerns and confusing points as listed below, which should be addressed before being considered for publication in IJMS.
Specific comments
(1) In the transgenic Arabidopsis lines, how does infection with P.parasitica or PstDC3000 (Fig.2) “induce” VaRGA1 expression from a constitutively overexpressing 35S promoter? According to the method section, VaRGA1 cds was directly placed under 35S promoter. Is there any post-transcriptional regulation such as mRNA turnover for VaRGA1 transcript? Or did the cloning include a cis element conferring the inducibility?
Response: The expression of VaRGA1 in transgenic lines increased after infection with P.parasitica or PstDC3000, which may be due to an increase in mRNA stability of VaRGA1 under pathogen stress. Further experiments need to be conducted to verify any post-transcriptional regulation and to study how it can be regulated.
(2) Although this study addresses new aspects of VaRGA1 function in JA-mediated defense and different types of diseases, the conclusion regarding VaRGA1 improve SA-mediated defense was already shown in the group’s previous publication (Li, X. et al. 2017) using a very similar overexpression approach using N. benthamiana. I feel the manuscript will greatly benefit from examining the VaRGA1’s function in a little more detail, taking advantage of genetic resources of Arabidopsis. For example, would VaRGA1 be able to improve disease resistance in mutants defective in SA/JA pathways? Or is it completely dependent on the hormone response pathways?
Response: The NBS-LRR gene with broad-spectrum resistance is rare. In our previous study, VaRGA1 was induced by Plasmopara viticola and could improve the resistance of tobacco to Phytophthora capsici. by activating SA signaling pathways (Li, X. et al. 2017). We suspect that VaRGA1 may have a broad spectrum of resistance to various pathogens including biotrophic P. viticola, hemi-biotrophic PstDC3000 and necrotrophic B. cinerea, and therefore, the resistance of VaRGA1 to different pathogenic bacteria and the corresponding hormone approach were studied. Your suggestion is helpful to our future research, and we will purchase the mutant seeds defective in SA/JA pathways in the next step, and to detect the resistance of the VaRGA1 transgenic line to the above-mentioned pathogenic bacteria, so as to confirm whether the disease-resistant function of VaRGA1 is independent of the corresponding pathways of the hormones.
(3) As examined in N. benthamiana in their previous publication, subcellular localization of the over-expressed VaRGA1 in the transgenic Arabidopsis lines should be examined. Does VaRGA1 localize in both nucleus and cytoplasm in Arabidopsis? Does disease infection change the localization pattern? Would excluding VaRGA1 from nucleus by a nuclear export signal abolish the observed effect of VaRGA1 in the transgenic lines?
Response: We observed that VaRGA1 was also located in nucleus and cytoplasm in Arabidopsis thaliana (Supplementary Figure 1). The subcellular localization of proteins plays an important role in plant immune response. Current studies have shown that a large number of NBS-LRR proteins are located in cytoplasm and nucleus, and the localization in the nucleus is the key to activating the defense response. Whether the localization of VaRGA1 is affected by the invasion of pathogenic bacteria and whether the transgenic plants still have disease resistance after VaRGA1 is transported out of the nucleus by nuclear output signal, is a topic worthy of study.
(4) The nature of “atrga1” mutant is not clear. There is no Arabidopsis gene with ID “ATG43098” (l.356). Please provide correct accession number and identification number for the “atrga1” mutant line. “homologous” should be explained as well. How about % identity and similarity to VaRGA1? Are there multiple genes with comparable similarity/identity? If so, how did the authors select one?
Response: We are very sorry about the ID of this gene and the correct identification number is “AT5G36930”. Two mutants of A. thaliana this gene called N638139 and N626884 were purchased from NASC. The homology of the protein sequence of AT5G36930 to VaRGA1 was 41.18%, which is the gene with the highest similarity with VaRGA1 in Arabidopsis thaliana. Among many homologous genes, AT5G36930 was the most similar gene in Arabidopsis and it matched the VaRGA1 protein sequence. Both of them have a TIR-NBS-LRR domain, which is a typical domain of resistance gene. So we selected this gene for further study.
(5) For histochemical staining data, mock treatment for each genotype should be included (Fig. 3C, 5C, 7C) to show the basal level of H2O2/cell death/O2- in each line.
Response: The experiment was performed in a previous pre-experiment. In the absence of pathogen treatment, the leaves stained with DAB and trypan blue were nearly transparent, and the leaves stained with aniline blue could only see a very small amount of callosum under microscope. When the formal experiment was carried out, the growth environment and treatment conditions of each group of materials were consistent as much as possible. Even so, your suggestion is scientific and rigorous and will be actively adopted in our future experiments.
(6) At 0-hpi, AtNPR1 and AtEDS1 expression show statistical difference among genotypes in some experiments but not in the others (Fig. 4,6,8). Is this due to different condition of 0-hpi plants for each pathogen? It would be worth clarifying a condition where the key hormone regulatory genes are altered even in the absence of biotic stresses.
Response: Due to the inconsistency of the best growth conditions of different pathogens, there are different requirements for the growth stage of inoculated materials and the need for in vitro vaccination. This is likely to make a difference in the expression of the corresponding genes. However, in the vaccination experiment of a certain strain, the consistency between different plant growth conditions and vaccination conditions was guaranteed as much as possible, so the data obtained were credible.
(7) Fig. 9 identifies potential cis-regulatory element candidates, but the search result alone without an empirical support is not a strong argument for their actual functions. In fact, VaRGA1 cds without any of these elements still show some induction upon infection (Fig.2). Authors can validate the importance (or lack thereof) of each candidate element by a reporter gene assay. Without that, I suggest making Fig.9 a supplemental figure.
Response: As shown in figure 1, VaRGA1 expression was strongly induced by SA and suppressed by JA in V. amurensis “Shuanghong”. Therefore, the promoter sequence of the gene was analyzed, and TCA-element involved in SA responsiveness were found. Further experiments need to be carried out to confirm whether each predicted candidate element works and looks for the core region of the promoter in response to the pathogens. The expression of VaRGA1 in transgenic lines increased after infection with P.parasitica or PstDC3000, which may be due to the increase of mRNA stability of VaRGA1 under pathogen stress.
(8) l.98: VaRGA1 expression is suppressed, not induced, by JA in Fig. 1.
Response: In general, the SA and JA signaling pathways inhibit each other. After the spraying of JA, the SA content of the plant decreased. As shown in figure 1, the expression of VaRGA1 was induced by SA. Therefore, VaRGA1 expression was suppressed by JA.
(9) l.89: [35] should be [34]? The reference [34] did not seem to mention about P. capsici.
Response: The reference [34] described the broad-spectrum resistance of RLM3 and cited it only once when explaining this view. The reference [35] is closely related to our study and has been cited several times in this paper.
We tried our best to improve the manuscript and made some changes in the manuscript.. These changes will not influence the content and framework of the paper. Thank you very much for your comments and suggestions.
Round 2
Reviewer 2 Report
The authors have made some changes which clarified results and methods in the text. However, some major concerns remain in the revised manuscript still, therefore it still requires careful revision before publication.
Response to author’s comment
(1) Please clarify and discuss the post-transcriptional regulation possibility in the text. I believe it would help readers understand the situation.
(3) Figure S1. Nuclear localization apparent only in stomata guard cells nad not clear in epidermal cells. Could authors incorporate DAPI staining to show clearer nuclear localization in epidermal cells?
(4) L.379- Text not revised. Please pay attention and incorporate these correct information in the text.
(5) These are controls and equally important in this manuscript as well. As for the basal level, it's ideal to show the "nearly transparent" leaf images - if it's difficult, at least mention that in the text. "Mock" differs from "before treatment". Mock treatments show how much signals arise from experimental artifacts caused by handling. I believe mock treatment as a control is important in the context of this manuscript. I request authors to perform mock treatment before resubmission.
(6) Please clarify this discussion in text.
(7) Authors did not address my concern here. In my opinion, the significance of the figure is very low without additional lines of support. If authors do NOT plan to examine any of the elements empirically in this manuscript, the figure should be toned down by making it as supplemental because there is no additional evidence indicating that these elements found are important or even functional for the VaRGA1 expression regulation. Yes there is biotic stress response in grapes but there is no evidence presented in this manuscript, regarding any of these elements are involved. In Arabidopsis overexpression lines without any of these elements, VaRGA1 was still induced by biotic stress. I strongly suggest making it to supplemental, or remove and save it for the future study where the authors characterize these cis-elements.
(8) The section title “2.1. VaRGA1 Expression is Induced by SA and JA in Grape (l.106)” says VaRGA1 is induced also by JA, which is simply wrong. Please correct this error.
Author Response
(1) Please clarify and discuss the post-transcriptional regulation possibility in the text. I believe it would help readers understand the situation.
Response: The post-transcriptional regulation possibility was discussed in the second paragraph of Discussion.
(3) Figure S1. Nuclear localization apparent only in stomata guard cells and not clear in epidermal cells. Could authors incorporate DAPI staining to show clearer nuclear localization in epidermal cells?
Response: The reason for this phenomenon was that the guard cells and epidermis cells were not on the same plane during observation, resulting in weak fluorescence in the epidermis cells. In fact, the subcellular localization of epidermis cells was consistent with that of guard cells. In order to prove the location of this gene in the epidermal cells, we provide a corresponding picture (Supplementary Figure 2B).
(4) L.379- Text not revised. Please pay attention and incorporate these correct information in the text.
Response: The corresponding information has been revised.
(5) These are controls and equally important in this manuscript as well. As for the basal level, it's ideal to show the "nearly transparent" leaf images - if it's difficult, at least mention that in the text. "Mock" differs from "before treatment". Mock treatments show how much signals arise from experimental artifacts caused by handling. I believe mock treatment as a control is important in the context of this manuscript. I request authors to perform mock treatment before resubmission.
Response: This experiment (including Mock) has been repeated many times before, and the results were consistent. Mock treatments of histochemical staining were shown in Supplementary Figure 3.
(6) Please clarify this discussion in text.
Response: The discussion about "nearly transparent" leaf images and Mock were shown in the fifth paragraph of Discussion.
(7) Authors did not address my concern here. In my opinion, the significance of the figure is very low without additional lines of support. If authors do NOT plan to examine any of the elements empirically in this manuscript, the figure should be toned down by making it as supplemental because there is no additional evidence indicating that these elements found are important or even functional for the VaRGA1 expression regulation. Yes there is biotic stress response in grapes but there is no evidence presented in this manuscript, regarding any of these elements are involved. In Arabidopsis overexpression lines without any of these elements, VaRGA1 was still induced by biotic stress. I strongly suggest making it to supplemental, or remove and save it for the future study where the authors characterize these cis-elements.
Response: We took your suggestion and made the Figure 9 to Supplementary Figure 4.
(8) The section title “2.1. VaRGA1 Expression is Induced by SA and JA in Grape (l.106)” says VaRGA1 is induced also by JA, which is simply wrong. Please correct this error.
Response: We are very sorry for this error and the title should be “VaRGA1 Expression is affected by SA and JA in Grape”.

Round 3
Reviewer 2 Report
The authors have addressed most of the concerns and improved the manuscript. I have two concerns left. In particular, given the relevance of the Arabidopsis mutant line ("atrga1") in this study, please consider the mutant identity a major concern. The mutant's identifiable information, as well as a few other information/discussion listed below, have to be made clear in the text before publication.
(4)-A The homology and similarity information you provided in the previous response should be incorporated in the text as well.
(4)-B The identity of mutant lines used in the study needs to be provided in the text. In the previous response, authors provided two NASC ids for atrga1 (AT5G36930); "N638139" and "N626884", which correspond to germplasms/ABRC stock SALK_138139 and SALK_126884, respectively. Insertion positions are quite different - one is in the gene body, the other is in the promoter. Which line is used in this study? SALK_138139 appears to have multiple insertions including another gene AT5G66030, according to ABRC (https://abrc.osu.edu/stocks/number/SALK_138139). Did author (or a previous study) confirm genotype/loss of function/expression of the AtRGA1 in the mutant line by PCR and/or RT-PCR? The mutant may be showing the expected phenotype, but a confirmation is necessary to link the phenotype to the gene of interest (AT5G36930 in this case). This is particularly important if the mutant line has never been characterized before.
(6) I meant the discussion regarding AtNPR1 and AtEDS1 expression at 0-hpi in authors previous response. It's copied below.
"Due to the inconsistency of the best growth conditions of different pathogens, there are different requirements for the growth stage of inoculated materials and the need for in vitro vaccination. This is likely to make a difference in the expression of the corresponding genes. However, in the vaccination experiment of a certain strain, the consistency between different plant growth conditions and vaccination conditions was guaranteed as much as possible, so the data obtained were credible."
Author Response
The authors have addressed most of the concerns and improved the manuscript. I have two concerns left. In particular, given the relevance of the Arabidopsis mutant line ("atrga1") in this study, please consider the mutant identity a major concern. The mutant's identifiable information, as well as a few other information/discussion listed below, have to be made clear in the text before publication.
(4)-A The homology and similarity information you provided in the previous response should be incorporated in the text as well.
Response: The corresponding information has been revised in Materials and Methods (4.1).
(4)-B The identity of mutant lines used in the study needs to be provided in the text. In the previous response, authors provided two NASC ids for atrga1 (AT5G36930); "N638139" and "N626884", which correspond to germplasms/ABRC stock SALK_138139 and SALK_126884, respectively. Insertion positions are quite different - one is in the gene body, the other is in the promoter. Which line is used in this study? SALK_138139 appears to have multiple insertions including another gene AT5G66030, according to ABRC (https://abrc.osu.edu/stocks/number/SALK_138139). Did author (or a previous study) confirm genotype/loss of function/expression of the AtRGA1 in the mutant line by PCR and/or RT-PCR? The mutant may be showing the expected phenotype, but a confirmation is necessary to link the phenotype to the gene of interest (AT5G36930 in this case). This is particularly important if the mutant line has never been characterized before.
Response: N626884 (atrga1) was used in this study and identified by PCR and RT-PCR (Supplementary Figure 2). Details were shown in Materials and Methods (4.1).
(6) I meant the discussion regarding AtNPR1 and AtEDS1 expression at 0-hpi in authors previous response. It's copied below.
"Due to the inconsistency of the best growth conditions of different pathogens, there are different requirements for the growth stage of inoculated materials and the need for in vitro vaccination. This is likely to make a difference in the expression of the corresponding genes. However, in the vaccination experiment of a certain strain, the consistency between different plant growth conditions and vaccination conditions was guaranteed as much as possible, so the data obtained were credible."
Response: The statistical difference about AtNPR1 and AtEDS1 expression among genotypes at 0-hpi was discussed in paragraph 6 of Discussion.
We tried our best to improve the manuscript and made some changes in the manuscript.. Thank you very much for your comments and suggestions.
